# Evidence for pressure induced unconventional quantum criticality in the coupled spin ladder antiferromagnet $C_9H_{18}N_2CuBr_4$

Tao Hong[1 ✉], Tao Ying[2], Qing Huang[3], Sachith E. Dissanayake[4], Yiming Qiu[5],
Mark M. Turnbull[6], Andrey A. Podlesnyak[1], Yan Wu[1], Huibo Cao[1], Yaohua Liu[1,7], Izuru Umehara[8],
Jun Gouchi[9], Yoshiya Uwatoko[9], Masaaki Matsuda[1], David A. Tennant[3,10], Gia-Wei Chern[11],
Kai P. Schmidt[12] & Stefan Wessel[13]

Quantum phase transitions in quantum matter occur at zero temperature between distinct ground states by tuning a nonthermal control parameter. Often, they can be accurately described within the Landau theory of phase transitions, similarly to conventional thermal phase transitions. However, this picture can break down under certain circumstances. Here, we present a comprehensive study of the effect of hydrostatic pressure on the magnetic structure and spin dynamics of the spin-1/2 ladder compound $C_9H_{18}N_2CuBr_4$. Single-crystal heat capacity and neutron diffraction measurements reveal that the Néel-ordered phase breaks down beyond a critical pressure of $P_c \sim 1.0$ GPa through a continuous quantum phase transition. Estimates of the critical exponents suggest that this transition may fall outside the traditional Landau paradigm. The inelastic neutron scattering spectra at 1.3 GPa are characterized by two well-separated gapped modes, including one continuum-like and another resolution-limited excitation in distinct scattering channels, which further indicates an exotic quantum-disordered phase above $P_c$.

[1] Neutron Scattering Division, Oak Ridge National Laboratory, Oak Ridge, TN 37831, USA. [2] School of Physics, Harbin Institute of Technology, 150001 Harbin, China. [3] Department of Physics and Astronomy, University of Tennessee, Knoxville, TN 37996, USA. [4] Department of Mechanical Engineering, University of Rochester, Rochester, NY 14617, USA. [5] National Institute of Standards and Technology, Gaithersburg, MD 20899, USA. [6] Carlson School of Chemistry and Biochemistry, Clark University, Worcester, MA 01610, USA. [7] Second Target Station, Oak Ridge National Laboratory, Oak Ridge, TN 37831, USA. [8] Department of Physics, Yokohama National University, Yokohama 240-8501, Japan. [9] Institute for Solid State Physics, University of Tokyo, 5-1-5 Kashiwanoha, Kashiwa, Chiba 277-8581, Japan. [10] Department of Materials Science and Engineering, University of Tennessee, Knoxville, TN 37996, USA. [11] Department of Physics, University of Virginia, Charlottesville, VA 22904, USA. [12] Lehrstuhl für Theoretische Physik I, Friedrich-Alexander-Universität (FAU) Erlangen-Nürnberg, Staudtstrasse 7, Erlangen D-91058, Germany. [13] Theoretische Festkörperphysik, JARA-FIT and JARA-HPC, RWTH Aachen University, 52056 Aachen, Germany. ✉email: hongt@ornl.gov

Landau's symmetry-breaking theory[1,2] forms a cornerstone for understanding many phases of matter in condensed matter physics. However, despite its remarkable success, various phenomena, such as the fractional quantum Hall effect[3,4] and topologically ordered quantum matter[5,6], have exposed the limitations of this paradigm. Moreover, continuous zero-temperature quantum phase transitions (QPTs), which exhibit emergent gauge fields and fractionalized (deconfined) degrees of freedom, also extend beyond this conventional framework. In recent years, several model systems, proposed to exhibit deconfined quantum criticality, have been exhaustively studied by analytical[7,8] and numerical[9–12] methods. However, to the best of our knowledge, thus far there has still been no experimental realization of such unconventional QPTs with fractionalization, and suitable experimental platforms are thus highly desirable.

Here, we focus on the $S = 1/2$ magnetic insulator $C_9H_{18}N_2CuBr_4$ (DLCB for short)[13]. Its magnetic properties at ambient pressure were examined in detail recently[14–16]. As seen from its crystal structure, shown in Fig. 1a, this compound is composed of coupled two-leg spin ladders, with the chain direction extending along the $b$-axis. The inter-ladder coupling in DLCB is sufficiently strong to drive the system into an anti-ferromagnetically ordered phase below 2.0 K[14]. The staggered moments point alternately along an easy axis ($\equiv \hat{z}$), the $c^*$-axis in the reciprocal space with an ordered moment size of 0.39(4) $\mu_B$, which is just 40% of the saturated moment size, due to strong quantum fluctuations that affect this material.

The magnetic excitations of DLCB at ambient pressure can be described quantitatively[15] by the Hamiltonian of a two-dimensional quantum spin model of coupled ladders for the magnetic interactions:

$$H = \sum_{\gamma, \langle i,j \rangle} J_\gamma \left[ S_i^z S_j^z + \lambda \left( S_i^x S_j^x + S_i^y S_j^y \right) \right],$$  (1)

where the subscript $\gamma$ reads either 'rung', 'leg', or 'int'—for $J_\gamma$ the

rung, leg, or interladder exchange constant—and $i$ and $j$ are nearest-neighbor lattice sites. The parameter $\lambda$ specifies an exchange anisotropy, with $\lambda = 0$ and 1 defining the limiting cases of Ising and Heisenberg interactions, respectively. To minimize the number of parameters to fit the experimental dispersions, $\lambda$ is assumed to be the same along all exchange paths[15]. This assumption is made to avoid overparameterization but does not affect the main conclusion regarding the properties of this model system.

Owing to an Ising-type anisotropy, $\lambda < 1$, the polarized neutron study[16] confirmed that the gapped triplet ($S = 1$ and $S_z = 0, \pm 1$) excitation energy splits into a gapped doublet ($S = 1$ and $S_z = \pm 1$) as the transverse mode (TM) and a gapped "singlet" ($S = 1$ and $S_z = 0$) as the longitudinal or amplitude mode (LM)[17–25], reflecting spin fluctuations perpendicular and parallel to the easy axis, respectively. Importantly, an analysis of the spin Hamiltonian suggests that DLCB is close to the quantum critical point (QCP) at ambient pressure and zero field[14–16,26], and thus its magnetic properties could be extraordinarily responsive to an external stimulus such as hydrostatic pressure.

In this work, we present compelling AC heat capacity and neutron scattering results on DLCB under applied hydrostatic pressure up to 1.7 GPa, which directly detect the magnetic order and dynamic structure factor (DSF) and thus allow us to address the nature of the static and dynamic spin correlations under pressure. The applied pressure is demonstrated to be a parameter that effectively tunes the exchange interactions in the spin Hamiltonian without inducing a structural transition. Figure 1b summarizes the phase diagram of DLCB as determined from this study as a function of temperature and pressure as well as the pressure dependence of anisotropic energy gaps. In the following, we will illustrate the collapse of the antiferromagnetic (AFM) ordering at $P_c \sim 1.0$ GPa through a continuous quantum phase transition. From our analysis, we furthermore obtain an estimate for the anomalous exponent $\eta$, which describes the spatial decay

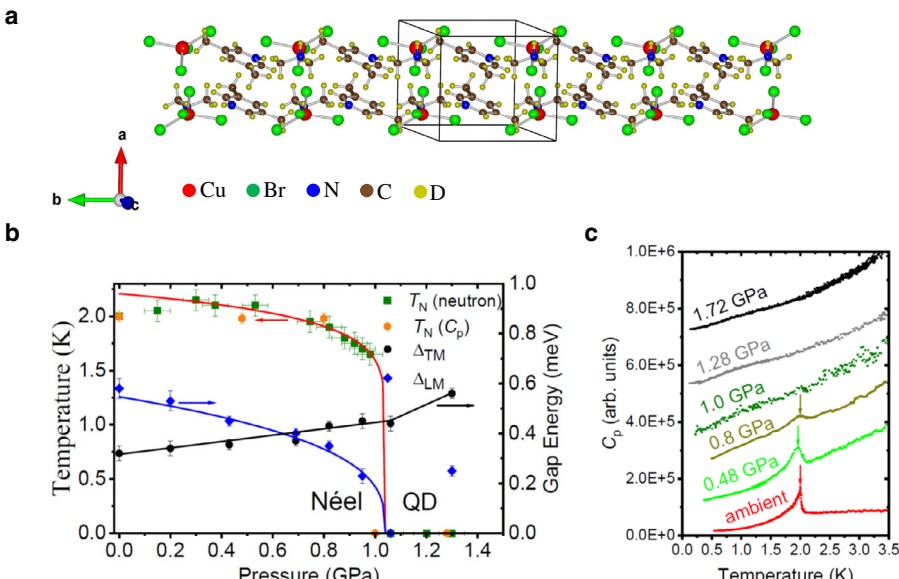

**Fig. 1 Crystal structure and phase diagram of $C_9H_{18}N_2CuBr_4$ (DLCB for short) as a function of pressure and temperature. a** Crystal structure of deuterated $C_9H_{18}N_2CuBr_4$ projected along the crystallographic $c$-axis to show the stacking of discrete DMA⁺ ($C_2D_8N$) and 35DMP⁺ ($C_7D_{10}N$) cations. Outlined is a nuclear unit cell. **b** Phase diagram of DLCB as a function of pressure and temperature, showing a transition between the Néel ordered phase and the exotic quantum-disordered (QD) phase as well as the pressure dependence of anisotropic energy gaps. Circle and diamond points are the energy gaps of the transverse mode (TM) and longitudinal mode (LM), respectively. The red and black solid lines are guides to the eye. The blue line was obtained from a fit of $\Delta_{LM}(P) \propto (P_c - P)^\nu$ with $P_c = 1.04$ GPa and the exponent $\nu = 0.33$. **c** The AC heat capacity $C_P$ of DLCB as a function of temperature at ambient pressure, 0.48, 0.8, 1.0, 1.28, and 1.72 GPa. For clarity, the data are shifted upwards. The transition temperature is indicated by an arrow. Error bars represent one standard deviation.

of the magnetic correlations, and that is unusually large at this QCP as compared to conventional values. Moreover, we observe broad magnetic excitation continua near the phase transition, which can not be attributed to the broadening effect of either chemical disorder or two-magnon scattering processes in DLCB. Rather, we propose that additional (frustrating) interlayer couplings, which become more pronounced close to the transition, may contribute to drive the system into an exotic gapped quantum paramagnetic regime, such as a quantum spin liquid phase[27–30].

## Results

**AC heat capacity and neutron diffraction results under pressure.** Figure 1c shows the AC heat capacity of a deuterated single crystal of DLCB as a function of temperature. At ambient pressure, a sharp anomaly indicates a phase transition to the AFM phase at $T_N = 2.0$ K. This peak becomes broader and smaller at 0.48 and 0.8 GPa, indicative of a suppression of the AFM order with increasing pressure. At and beyond 1.0 GPa it becomes indiscernible at least down to temperatures of 0.15 K. To explore this collapse of the Néel order in more detail, we measured the order parameter of DLCB using single-crystal neutron diffraction. Figure 2a, b shows the pressure-dependence of neutron diffraction $\theta/2\theta$ scans at the AFM wavevector $\mathbf{q} = (0.5, 0.5, -0.5)$. The scattering intensity of the magnetic Bragg peak becomes diminished as hydrostatic pressure increases. At and above a hydrostatic pressure of 1.05 GPa, there is no experimental evidence of magnetic order down to 0.25 K. Moreover, no evidence is found of any incommensurate magnetic Bragg peak or diffuse scattering over a wide range of reciprocal space at $T = 0.25$ K and $P = 1.3$ GPa in Fig. 2c. The refinement of additional single-crystal

neutron diffraction data at and above 1.05 GPa does not reveal any structural distortion as the possible cause for the absence of magnetic order and the triclinic space group P$\bar{1}$ is preserved at each pressure, as listed in Supplementary Table 1. The temperature-dependent order parameter has been measured at various pressures as well. Figure 2d outlines the determined pressure dependence of the ordered moment size $m$ that is continuously reduced to 0.07(5) $\mu_B$ at 0.98 GPa, indicating a continuous QPT[31,32]. The best fit to $m \propto (P_c - P)^\beta$ yields $P_c = 1.04(4)$ GPa and the order-parameter exponent $\beta = 0.68(5)$. This value is appreciably larger than, e.g., the values of $\beta \simeq 0.33$ and $\beta = 0.5$ for the (2+1)D and (3+1)D Ising universality classes[33], respectively (an analysis of the scaling behavior over the entire investigated pressure region is provided in Supplementary Note 2). It is noteworthy that the determined AFM ordering temperature $T_N$ increases slightly upon initially increasing the pressure and then decreases gradually with pressure until its abrupt drop towards zero at $P_c$. This behavior is rather distinct from the behavior near a conventional QCP, where $T_N$ exhibits a smooth power-law decrease towards zero at the QCP. We take this as an indication that the underlying physics is indeed distinct from a conventional order-to-quantum disorder transition scenario.

**Inelastic neutron scattering results under pressure.** To gain insight into the nature of the pressure-induced quantum-disordered (QD) phase, inelastic neutron scattering has been used to probe the evolution of the dynamic spin correlation function of DLCB as a function of pressure. Figure 3a, b shows background-subtracted energy scans at the same AFM wavevector for various pressures. The spectral lineshapes for $P < P_c$ (or $P > P_c$) are spin-

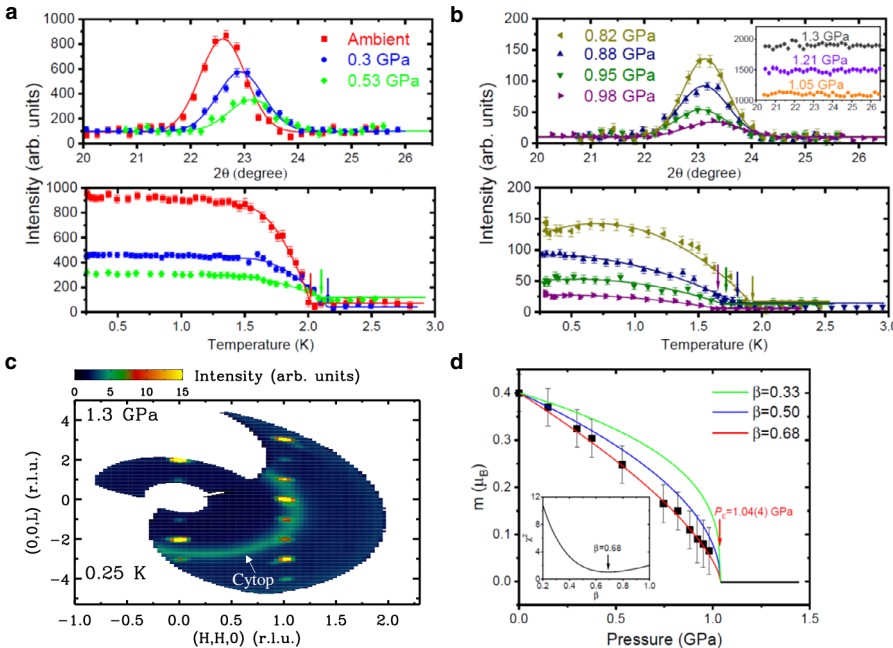

**Fig. 2 Single-crystal neutron diffraction study under pressure.** The representative $\theta/2\theta$ scans and the temperature dependence of the neutron scattering intensity measured at a cold neutron triple-axis spectrometer (CTAX) around the antiferromagnetic wavevector $\mathbf{q} = (0.5, 0.5, -0.5)$ for (**a**) ambient pressure, 0.3 and 0.53 GPa and for (**b**) 0.82, 0.88, 0.95, and 0.98 GPa, respectively. For $\theta/2\theta$ scans, the solid lines are fits to the Gaussian profile. For clarity, the data above 1 GPa presented in the inset are shifted vertically by successive increments of 400. For temperature dependence of the order parameter, the transition temperature at each pressure is indicated by the arrow and the solid lines are guides to the eye. Experimental data were collected at 0.25 K. **c** Single-crystal neutron diffraction pattern measured at a cold neutron chopper spectrometer (CNCS) at $T = 0.25$ K and $P = 1.3$ GPa. The ring feature originates from the cytop glue. **d** Ordered moment $m$ as a function of applied pressure, along with fits to $m(P) \propto (P_c - P)^\beta$ for several typical values of $\beta$. The best fit results for $\beta = 0.68(5)$ and $P_c = 1.04(4)$ GPa. The inset shows the goodness of the fit for different values of $\beta$. Error bars represent one standard deviation.

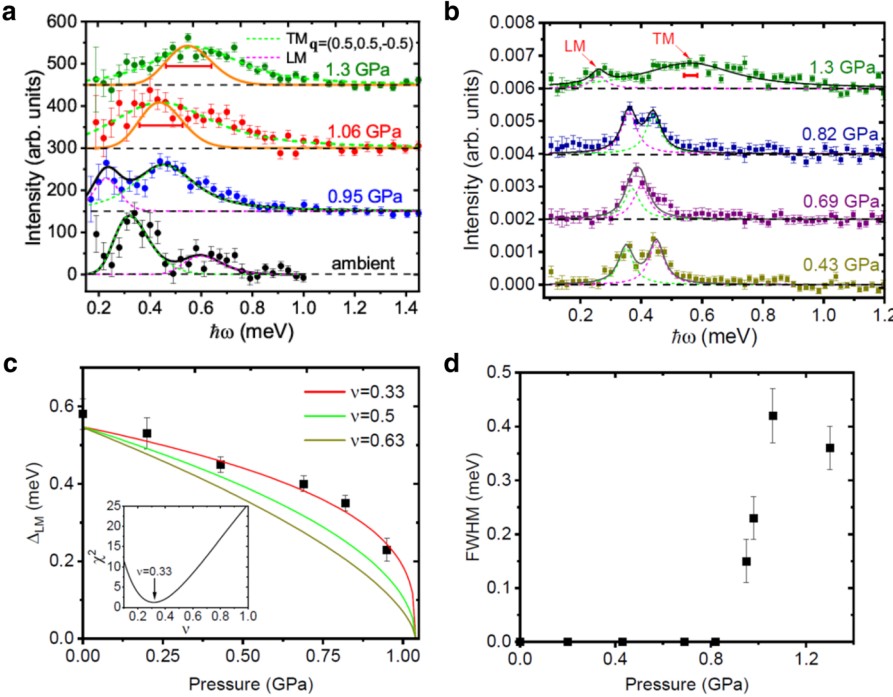

**Fig. 3 Single-crystal inelastic neutron scattering study under pressure.** The representative background-subtracted transferred energy scans measured (**a**) at a high intensity multi-axis crystal spectrometer (MACS) and (**b**) at CNCS at the antiferromagnetic wavevector $\mathbf{q} = (0.5, 0.5, -0.5)$ for several pressures. All data were collect at $T = 1.5$ K except that the data at 1.3 GPa (CNCS) were measured at 0.25 K. For clarity, the data and fits are shifted vertically by successive increments of 150 (MACS) or 0.002 (CNCS). The green and magenta dashed lines are fits to the damped harmonic-oscillator (DHO) model convolved with the instrumental resolution for the TM and LM, respectively. The black solid lines are their sum. The green and orange solid lines at 1.06 and 1.3 GPa (MACS) are convolved calculations for the TM by the DHO model and quantum Monte Carlo (QMC) calculations, respectively. The black dashed lines are guides to the eye. The red horizontal bars represent the instrumental resolution. **c** The pressure dependence of $\Delta_{LM}$ below $P_c$, along with fits to $\Delta_{LM}(P) \propto (P_c - P)^{\nu z}$, for different values of $\nu$ with $z = 1$. The best fit gives $\nu = 0.33(4)$. The inset shows the goodness of the fit for different values of $\nu$. **d** The pressure dependence of the intrinsic linewidth of the TM damping. Error bars represent one standard deviation.

gapped and were modeled by a superposition of two (or single) double-Lorentzian damped harmonic-oscillator (DHO) modes, convolved with the instrumental resolution function. At ambient pressure, the best fit yields the gap energies of the TM and LM as $\Delta_{TM} = 0.32(3)$ meV and $\Delta_{LM} = 0.58(4)$ meV, respectively. Their values are consistent with the previously reported values[16]. We find that the peak profiles of both the TM and LM for $P \leq 0.82$ GPa are limited by the instrumental resolution. Notably, the TM becomes broad at 0.95 GPa with an intrinsic full width at half maximum (FWHM) of 0.15(4) meV. Figure 1b summarizes the pressure dependence of the extracted excitation energies $\Delta_{TM}$ and $\Delta_{LM}$ across the quantum phase transition. The slightly growing gap energy of $\Delta_{TM}$ can delay the thermal depletion of the magnetic order towards higher $T_N$. The LM gap $\Delta_{LM}$ softens along with the decrease of the ordered moment and the best fit to $\Delta_{LM} \propto (P_c - P)^{\nu z}$ in Fig. 3c, where $\nu$ is the correlation-length exponent and $z$ is the dynamic critical exponent, was determined. Assuming $z = 1$ at the pressure-induced quantum phase transition[31], we obtain $\nu = 0.33(4)$ which is smaller than the values of $\nu \simeq 0.63$ and $\nu = 0.5$ for the (2+1)D and (3+1)D Ising universality classes[33], respectively. However, caution should be used when interpreting this finding as the density of data available near $P_c$ is barely sufficient for an accurate estimate. Notably, recent quantum Monte Carlo (QMC) work finds a similarly small value of $\nu \approx 0.45$ at the considered deconfined QCP in the $S = 1/2$ square-lattice $J$-$Q$ model[34,35]. At 1.06 GPa, slightly above $P_c$, the LM has a very small gap that cannot be distinguished within the limited instrumental resolution, while $\Delta_{TM}$ moves further up to 0.44(3) meV. Based on the spin Hamiltonian at ambient pressure in Eq. (1), the ground state in the QD phase is expected to be a

trivial dimerized quantum paramagnet, where spins are paired up into singlets, arranged in a regular pattern. Such a phase has sharp, gapped $S = 1$ spin-wave excitations. Surprisingly, the spectral line shape of the TM signal with a long tail extending up to 1.1 meV becomes significantly broader than the instrumental resolution and the best fit to the same double-Lorentzian DHO model gives an intrinsic linewidth with a FWHM of 0.42(5) meV. It is important to emphasize that the two-TM scattering processes is not allowed in the same energy regime as a consequence of the energy-momentum conservation rule. Such spectral broadening persists at least up to 1.3 GPa, i.e., well beyond the QCP, where the FWHM becomes 0.34(5) meV.

**Quantum Monte Carlo simulations.** To quantitatively compare the observed spin dynamics with theoretical models, we examine the DSF using large scale QMC calculations for the original spin Hamiltonian in Eq. (1). The Hamiltonian parameters at ambient pressure were previously obtained to best match the experimentally observed magnetic dispersions as $J_{leg} = 0.62$ meV, $J_{rung} = 0.66$ meV, $J_{int} = 0.20$ meV and $\lambda = 0.87$[15]. We establish that the applied pressure effectively tunes the exchange coupling ratio $\alpha$ between the inter-ladder and intra-ladder couplings, assuming no impact on the interaction anisotropy $\lambda$ and the ratio between $J_{rung}$ and $J_{leg}$ (cf. Supplementary Table 7 for further details). At the critical pressure and for 1.3 GPa, $\alpha$ is reduced to 0.14 and 0.05, respectively, from its value of 0.32 at ambient pressure. We find that the parameter set ($J_{leg} = 0.91$ meV, $J_{rung} = 0.97$ meV and $J_{int} = 0.13$ meV) or ($J_{leg} = 1.03$ meV, $J_{rung} = 1.1$ meV and $J_{int} = 0.05$ meV) gives good agreement with the experimental value

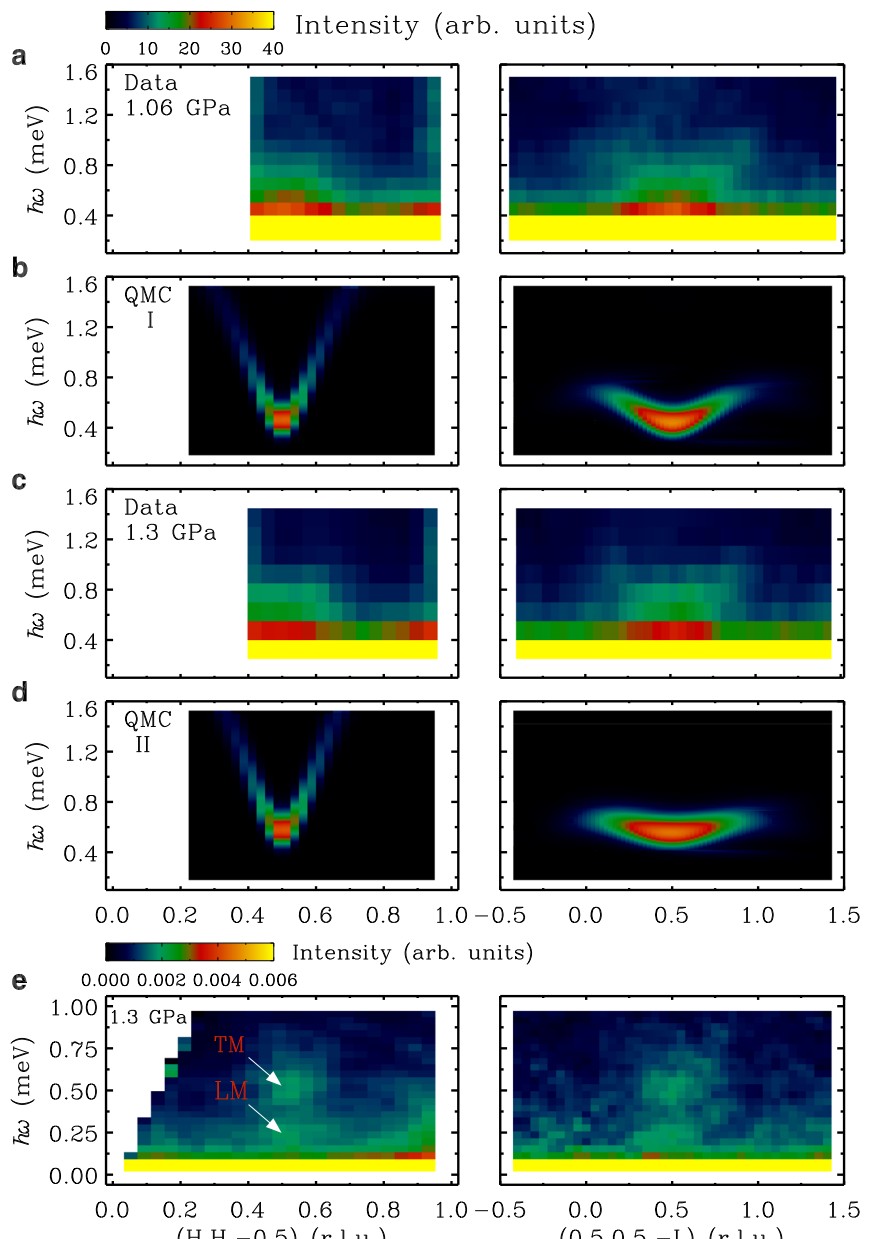

**Fig. 4 Comparison between the experimental data and QMC calculations.** False-color maps of the excitation spectra as a function of energy and wavevector transfer along two high-symmetry directions (H,H,−0.5) and (0.5,0.5,−L) in the reciprocal space, respectively. Experimental data were collected at MACS at $T = 1.5$ K and (**a**) $P = 1.06$ GPa and (**c**) $P = 1.3$ GPa. Data for H less than 0.4 r.l.u. are not shown due to a contamination by the direct neutron beam. **b**, **d** are dynamic structure factors of the TM calculated by QMC using the parameter sets at the critical point and 1.3 GPa, respectively, as described in the text. Simulations were convolved with the instrumental resolution function where the neutron polarization factor and the magnetic form factor for $Cu^{2+}$ were included. **e** High-resolution inelastic neutron scattering measurements at CNCS at $T = 0.25$ K and $P = 1.3$ GPa. The excitation spectra at $T = 15$ K are shown in Supplementary Fig. 6 (**e**). No smoothing or symmetrization was applied to all experimental data.

of $\Delta_{TM}$. As expected, the QMC calculations for the spin Hamiltonian in Eq. (1) with the best fit parameters for a pressure of 1.3 GPa indeed predict a trivial quantum paramagnetic ground state with conventional $S = 1$ spin-wave excitations. The calculated spectral lineshape profiles as indicated by the orange lines in Fig. 3a are visibly limited by the instrumental resolution, which are certainly a poor representation of the observed TM signal. Moreover, Fig. 4 shows the comparison over the entire Brillouin zone between the experimental excitation spectra at 1.06 and 1.3 GPa and the DSF as calculated by QMC, convolved with the instrumental resolution function. Clearly, the experimental data cannot be reproduced by the DSF calculated from the Hamiltonian in Eq. (1), which suggests that

the ground state above $P_c$ is potentially a more unconventional quantum disordered state.

**Discussion**

So what is the probable origin of the continuum-like broad excitations of DLCB? We have carried out single-crystal neutron diffraction experiments under pressure by measuring more than 300 nuclear Bragg peaks and the refinement results as listed in Supplementary Tables 2–5 do not reveal any evidence of chemical disorder under pressure except a small H/D isotope effect on the site occupancy. The latter has no influence on the magnetism of non-hydrogen-bonded systems such as DLCB. Consequently, it

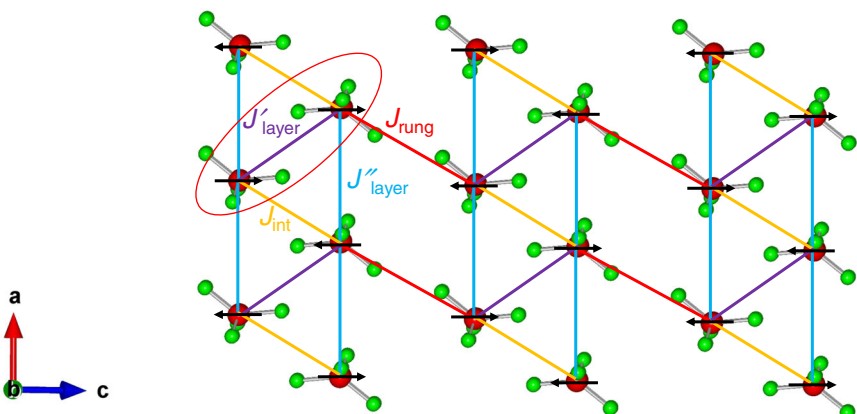

**Fig. 5 The magnetic interactions between Cu$^{2+}$ ions in the crystallographic *ac* plane.** Projection of (CuBr$_4$)$^{2-}$ tetrahedra onto a plane perpendicular to the *b*-axis. The organic cations play no role in the magnetism and are not shown. Black arrows indicate the directions of the spins in the Néel ordered phase. Red, orange, purple and light blue lines indicate the intraladder coupling $J_{rung}$, interladder coupling $J_{int}$, interlayer couplings $J'_{layer}$ and $J''_{layer}$, respectively. The ellipse outlines the frustrating inter-layer coupling $J'_{layer}$.

can be ruled out that the observed broadening is due to the chemical disorder. The other possible broadening effect attributed to spontaneous magnon decays[36–40] is also discussed in Supplementary Note 3 and can be excluded mainly due to violation of the kinematic conditions[41].

Another possible origin of the broad, continuum-like spectra observed in DLCB is fractionalization of the magnetic excitations i.e., spin-1/2 spinons, which are created and detected in pairs by neutron scattering as evident from conservation of the angular momentum. In such a scenario, the magnetic order parameter becomes a composite operator of partons, which should thereby lead to a large value of the anomalous exponent $\eta$[12], characterizing the spatial decay of the magnetic correlations. We can estimate $\eta$ based on the $\beta$ and $\nu$ values using the scaling relation[33] $\eta = 2\beta/\nu - (d + z - 2)$, where $d$ denotes the spatial dimension. Taking $d = 3$, based on the strong three-dimensional character of the magnetism at the transition, as we will discuss later, we obtain $\eta = 2.1(5)$ from $\beta = 0.68(5)$, $\nu = 0.33(4)$, and $z = 1$ as already stated above. This is in marked contrast to, e.g., the value $\eta = 0$ at the conventional (3+1)D Ising transition[33]. Also for $d = 2$, the value obtained for $\eta$ is significantly larger than, e.g., the value $\eta \approx 0.036$ at the conventional (2+1)D Ising transition[33]. These estimates for $\eta$, based on the critical exponents $\beta$ and $\nu$ and a (standard) scaling relation, may not represent the true asymptotic value in the quantum critical regime, due to the limitation of instrumental resolution. However, even in view of these caveats, the estimated values of $\eta$ are certainly significantly larger than the prediction at conventional QCPs or its mean-field approximation ($\eta = 0$). Such large values for $\eta$, along with the broad spectral linewidth observed near the quantum phase transition, as shown in Fig. 3d, suggest that the emergence of fractionalized excitations through the pressure-induced quantum phase transition may be realized in DLCB at the critical pressure.

Since DLCB is composed of interacting spin-1/2 ladders, it is also instructive to consider the one-dimensional (1D) limit of isolated ladders. Indeed, a 1D analog of the deconfined QCP (DQCP)[7,8] in an $S = 1/2$ chain system between the Ising ferromagnetic and valence bond solid (VBS) symmetry-breaking phases has been investigated recently[42,43]. Because higher-order four-spin interactions such as those around plaquettes are expected to be negligible for molecular spin-1/2 ladder systems, a QPT between the rung singlet (RS) and Néel phase emerges upon varying the Ising exchange anisotropy[44]. The ground state of the RS phase does not break any symmetry, and hence the RS-Néel transition is not a (1+1)D DQCP, but instead belongs to the

conventional (1+1)D Ising universality class. Furthermore, spin excitations in the VBS state fractionalize only at the DQCP, whereas the spinons become confined again in the disordered phase. In contrast, our high resolution excitation spectra in the QD regime at $P = 1.3$ GPa in Fig. 4e clearly show two well-separated gapped modes including one broad continuum-like and another almost dispersionless excitation. At the AFM wavevector, cf. Fig. 3b, we identify two distinct spin-gapped excitations: (i) a broad peak for the TM at about 0.56(4) meV with an intrinsic linewidth FWHM of 0.36(4) meV (i.e., five times broader than the instrumental resolution), and (ii) an additional, resolution-limited excitation for the LM at 0.25(3) meV. As a result, both conventional (1+1)D Ising quantum criticality, as well as a 1D DQCP in DLCB, can be excluded.

Since an ordinary quantum phase transition to a trivial quantum paramagnet emerges for unfrustrated interactions[31], as we observed also explicitly in the QMC simulations, a mechanism that is not included in the original Hamiltonian is required. The possibility of diagonal frustrating couplings in the ladders can be eliminated due to their unfavorable superexchange paths in DLCB. Space symmetry of the DLCB lattice also places strong restriction on the possible anisotropic Dzyaloshinskii-Moriya (DM) interactions. DM interactions in DLCB are forbidden along the rungs of the ladders and between the ladders by inversion symmetry, but are allowed along the ladder legs[45–48]. In general, DM interactions prefer non-collinear magnetic structures, and in DLCB the effects of the DM interactions on the collinear ordered state within the Néel phase should thus be very weak. Indeed, the field dependence of the anisotropic energy gaps at ambient pressure[16] agrees well with the Zeeman spectral splitting, which suggests that $S^z$ is a good quantum number. One may furthermore consider the couplings $J_{layer}$ between the two-dimensional layers. Their corresponding Cu$\cdots$Cu separation distances become reduced under pressure (see Supplementary Table 6 for further details) and may thus indeed become more pronounced in the vicinity of the critical point. Figure 5 shows the magnetic interactions between Cu$^{2+}$ ions in the crystallographic *ac* plane, in which the $J_{int}$, $J'_{layer}$ and $J''_{layer}$ interactions form triangular spin arrangements. In terms of the Cu-Br$\cdots$Br bridging angle, as summed up by the Goodenough–Kanamori–Anderson rules[49–51], the larger the deviation from 180°, the weaker is the anti-ferromagnetic superexchange and the coupling eventually becomes ferromagnetic for bridging angles close to 90°. Consequently, $J_{int}$, $J'_{layer}$ and $J''_{layer}$ are all antiferromagnetic exchange

interactions. While the coupling $J''_{layer}$ along the $a$-axis helps to align the ordered moments antiparallel to each other between adjacent layers in the Néel ordered phase, an enhancement of the coupling $J'_{layer}$ can lead to magnetic frustration. One possibility is that the system is driven into an exotic gapped quantum spin liquid phase in terms of a three-dimensional frustrated network of quantum spins with triangular motifs. Further theoretical work is certainly necessary to assess such a scenario.

In summary, we have performed AC heat capacity and neutron scattering experiments under pressure on the compound $C_9H_{18}N_2CuBr_4$ at low temperature and continuously tuned the ground state from the AFM Néel phase across a continuous quantum phase transition into a quantum disordered regime. The unique ability of neutron scattering to explore the dynamical spin correlation functions allows us to experimentally probe for fractionalized excitations. We indeed observe rather broad continuum-like magnetic excitation spectra near the transition, which—along with a large estimated value for the anomalous exponent $\eta$—suggests an interpretation in terms of fractionalized magnetic excitations. Our study, therefore, offers an interesting experimental platform to search for physics associated with unconventional quantum criticality and exotic quantum-disordered phases.

## Methods

**Single crystal growth**. Deuterated single crystals were grown using a solution method[13]. An aqueous solution containing a 1:1:1 ratio of deuterated (DMA)Br, (35DMP)Br, where DMA$^+$ is the dimethylammonium cation and 35DMP$^+$ is the 3,5-dimethylpyridinium cation, and the corresponding copper(II) halide salt was allowed to evaporate for several weeks; a few drops of DBr were added to the solution to avoid hydrolysis of the Cu(II) ion.

**AC heat capacity**. High-pressure experiment of AC heat capacity on a deuterated single-crystal of DLCB was performed in a dilution refrigerator using a Ni–Cr–Al hybrid piston-cylinder-type pressure cell[52]. Daphne oil 7373 was used as pressure-transmitting medium[53] and the pressure at low temperature was calibrated by measuring the superconducting transition of Pb[54]. The AC heat capacity $C_P$, which is inversely proportional to the observed lock-in voltages, was determined from the temperature modulation with a thermocouple. The technical details can be found in ref. [55]. Note that the AC heat capacity at ambient pressure was measured without the pressure cell and the observed almost linear-$T$ dependence above 2 K at finite pressures is caused by the background from the pressure-transmitting medium and the pressure cell.

**Neutron scattering measurements**. Single-crystal neutron diffraction measurements under pressure were carried out on a cold neutron triple-axis spectrometer (CTAX) with both the incident and final neutron energies fixed at 4.5 meV at High Flux Isotope Reactor (HFIR), Oak Ridge National Laboratory (ORNL). A standard helium-flow cryostat or helium-3 insert was used to achieve the base temperature of 1.45 K or 0.25 K. The nuclear structure at 5 K under pressure was determined using a closed-cycle refrigerator at the four-circle neutron diffractometer (HB-3A) at HFIR, ORNL with the incident wavelength of 1.55 Å. Elastic neutron scattering measurements were also performed using the CORELLI diffuse scattering spectrometer[56] at the Spallation Neutron Source (SNS), ORNL. A dilution refrigerator insert was used to reach the base temperature of 45 mK. The sample used in neutron diffraction measurements consists of one deuterated single crystal with mass of 50 mg and a 1.0° mosaic spread. Inelastic neutron scattering measurements under pressure using a standard helium-flow cryostat were performed on a high intensity multi-axis crystal spectrometer (MACS)[57] at the NIST Center for Neutron Research. The final neutron energy was fixed at 3.0 meV and the energy resolution at the elastic line is 0.13 meV. Inelastic neutron scattering data were also collected on a cold neutron chopper spectrometer (CNCS)[58] at SNS, ORNL. The scattering intensity was normalized to the number of incident protons per pulse and integrated out of the scattering plane direction by a narrow slice of ±0.1 r.l.u. in order to analyze the experimental data taken in the scattering plane. The experiment was performed using a helium-3 insert with the incident neutron energy of 2.07 meV and the energy resolution at the elastic line is 0.08 meV. The background was determined at $T = 15$ K under the same instrumental configurations. The sample used in inelastic neutron scattering consists of three co-aligned deuterated single crystals with a total mass of 200 mg and a 2.0° mosaic spread. Neutron scattering intensity as shown in Fig. 4 of the main text was integrated along the H or L directions by ±0.1 r.l.u. In all experiments, the sample was oriented in the (HHL) scattering plane and loaded inside a CuBe piston-cylinder-type pressure cell with maximum allowable hydrostatic pressure ~1.3 GPa. Fluorinert FC-770 was used as

pressure-transmitting medium to achieve good hydrostaticity. The desired hydrostatic pressure was applied by a hydraulic press. Change of volume with pressure in NaCl was used for pressure calibration with an accuracy of 0.1 GPa.

**Data analysis**. By assuming a collinear and commensurate antiferromagnetic structure, we can determine the ordered moment size under pressure $m(p)$ by normalizing the magnetic peak to the nuclear peak as:

$$m(p) = m_0 \sqrt{\frac{I(p)|_{mag}/I(p)|_{nuc}}{I_0|_{mag}/I_0|_{nuc}}}, \qquad (2)$$

where $m_0$ is the staggered moment size at ambient pressure. $I_0|_{mag}$ and $I(p)|_{mag}$ are the integrated intensities of the magnetic Bragg peak (0.5, 0.5, −0.5) at ambient pressure and under pressure, respectively. $I_0|_{nuc}$ and $I(p)|_{nuc}$ are the integrated intensities of the nuclear Bragg peak (0,0,2) at ambient pressure and under pressure, respectively.

The spectral lineshapes in Fig. 3 of the main text were fitted to the following double-Lorentzian DHO model[59,60]:

$$S(\hbar\omega) = \frac{A}{1 - \exp(-\hbar\omega/k_B T)}$$
$$\left[ \frac{\Gamma}{(\hbar\omega - \Delta)^2 + \Gamma^2} - \frac{\Gamma}{(\hbar\omega + \Delta)^2 + \Gamma^2} \right], \qquad (3)$$

where A is a prefactor, $k_B$ is the Boltzmann constant, $\Delta$ is the peak position and $\Gamma$ is the resolution-corrected intrinsic excitation linewidth i.e., half-width at half-maximum (HWFM) and convolved with the instrumental resolution function. The experimental resolution was calculated using the Reslib software[61].

**Quantum Monte Carlo calculations**. Quantum Monte Carlo (QMC) calculations were performed for the effective spin model for DLCB, based on the methods of ref. [15]. We used a combination of stochastic series expansion QMC simulations[62–64] and analytical continuations using the stochastic sampling scheme of ref. [65] in order to access the frequency-dependent spectral functions from imaginary-time correlation functions that are measured in the QMC calculations. From this approach, we obtain the spectral functions for the longitudinal channel, $S_L(\mathbf{k}, \omega) = \int dt \, e^{-ih\omega t} \langle S_{\mathbf{k}}^z(t) S_{-\mathbf{k}}^z(0) \rangle$, as well as for the transverse channel, $S_T(\mathbf{k}, \omega) = \int dt \, e^{-ih\omega t} \langle S_{\mathbf{k}}^+(t) S_{-\mathbf{k}}^-(0) + S_{\mathbf{k},}^-(t) S_{-\mathbf{k}}^+(0) \rangle$ as functions of momentum and frequency.

## Data availability

All data that support the findings of this study are available from the corresponding author upon reasonable request. Source data are provided with this paper.

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

## Acknowledgements

We gratefully acknowledge the insightful discussions with Ian Affleck, Collin Broholm, Hongcheng Jiang, Masashige Matsumoto, Cenke Xu, and Guangyong Xu. We also thank Matthew Collins, Michael Cox, Saad Elorfi, Cory Fletcher, Rick Goyette, Juscelino Leão, Christopher Redmon, Todd Sherline, Christopher Schmitt, Randall Sexton, Erik Stringfellow, and Tyler White for the technical support in neutron scattering experiments. A portion of this research used resources at the High Flux Isotope Reactor and Spallation Neutron Source, a DOE Office of Science User Facility operated by the Oak Ridge National Laboratory (ORNL). A portion of this research used resources of the Spallation Neutron Source Second Target Station Project at ORNL. Access to MACS was provided by the Center for High Resolution Neutron Scattering, a partnership between NIST and NSF under Agreement No. DMR-1508249. The research is also supported by Go! Student Program of ORNL.

## Author contributions

T.H. conceived the project. M.M.T. prepared the samples. I.U., G.J., and Y.U. measured the AC heat capacity. T.H., Q.H., S.E.D., Y.Q., A.P., M.M., Y.W., H.C., and Y.L. carried out the neutron-scattering measurements. T.Y. and S.W. performed the quantum Monte Carlo calculations. T.H., Q.H., D.A.T., G.-W.C., K.P.S., and S.W. analyzed the experimental data. T.H. wrote the manuscript with contributions from all co-authors.

## Competing interests

The authors declare no competing interests.
