## [Peer Review File · Nature Communications]

REVIEWER COMMENTS

Reviewer #1 (Remarks to the Author):

In this manuscript, the authors present their comprehensive experimental investigation of the organic magnet DLCB, which is a model $S=1/2$ spin ladder compound. While an isolated ladder has a singlet ground state with a large spin gap, when they are coupled with a large enough interladder coupling, they go into an ordered phase. DLCB is believed to lie just above this quantum critical point with weak magnetic order. The main idea of this work is to tune the magnetic Hamiltonian with hydrostatic pressure and study this quantum phase transition using elastic and inelastic neutron scattering. The authors found that the magnetic order is suppressed with moderate pressure of about 1 GPa. The first important observation is that this quantum phase transition is of second-order nature, which is shown by the continuous suppression of ordered moment size with increasing pressure. From this measurement, they extract the order parameter critical exponent β of 0.68, which is much larger than the value expected for a typical magnetic transition. In addition, the suppression of the longitudinal mode as a function of pressure exhibits power-law behavior with unusually small critical exponent ν . The second observation is the fact that the magnetic excitation spectrum in the phase $P > P_c$ cannot be explained by a conventional picture. If this quantum paramagnet is described by the conventional ladder picture (as assumed in the quantum Monte Carlo study), one would expect a well-defined excitation associated with the singlet-triplet gap. However, the observed spectrum does not show any sharp features and only a rather broad continuum was observed. The authors have carried out a careful examination to rule out any disorder effect for this continuum. The authors go on to argue that this quantum paramagnetic phase is a Z2 quantum spin liquid with fractionalized excitations, based on the 1.3 GPa spectrum, which seems to show a low-energy vison mode and spinon continuum excitations.

I find this manuscript very interesting and well-written. I think this work will be a valuable addition to the growing list of materials exhibiting quantum phase transitions. In particular, the careful experimental examination of the second-order nature of the transition is impressive and convincing. I recommend publication in Nat. Comm. After the authors address the following concerns:

1) While I am impressed with the experimental data and presentation about the critical behavior, the experimental evidence for fractionalization is weak. I do agree that the quantum phase transition in this compound seems to be quite unusual. However, I would expect to see much stronger evidence to make the logical jump from that to Z2 quantum spin liquid. In particular, the vison observation claim relies on only one scan done at the CNCS. The data measured at the same pressure at MACS do not seem to show similar structure. I recommend toning down the fractionalization claim.

2) Somewhat related comment: I am concerned with seemingly very different spectra from the two instruments. The raw spectra in the Supplementary material do not help either.

3) The QMC model calculation was done to show that one specific model for the quantum phase transition (variation of the interladder coupling α) fails to explain the observation. It is not clear to me why just varying α is the best model description.

4) The authors go on to discuss various possible microscopic origin for the observed unusual quantum phase transition, including discussions about the DM interactions. However, to me, this discussion is somewhat hollow, since the original Hamiltonian the authors consider, which treats the Ising anisotropy very phenomenologically. That is, λ in the Hamiltonian is assumed to be the same for all three interactions without microscopic justification.

Reviewer #2 (Remarks to the Author):

The authors have carried out an impressive study of the heat capacity, magnetic ordering, and magnetic excitations as a function of temperature and pressure in an $S=1/2$ spin-ladder material which they label DLCB. Notably, at a pressure of about 1.0 GPa there is a crossover from Neel order to a quantum spin liquid (QSL) phase. Much of the paper is devoted to speculation about the nature of the QSL phase; specifically, the authors argue that it is a gapped QSL with fractionalized degrees of freedom. Indeed, they claim to be able to identify the responses from vison and spinon excitations in the inelastic neutron spectra. Unfortunately, this model for the QSL seems to rely critically on the numerical values of the critical exponents β and ν which are determined from data entirely outside of the critical region. Typically, critical exponents are determined from high precision data in the reduced temperature range from 0.1 to 0.001 or 0.0001. Further, the data are analyzed with power laws modified by corrections-to-scaling and the effects of any crossover in universality class. By contrast, the power laws obtained in this paper are the results of fits of the data to simple power laws extending over a broad temperature range beginning around 0.1 and extending down to low temperatures, that is, entirely outside of the critical region. Thus they represent a parameterization of the low to intermediate temperature range data and have nothing to do with critical phenomena and actual critical exponents. This has led the authors to a vast overreach in the interpretation of their otherwise beautiful data. In reality, as far as I can tell, there is no evidence whatsoever that DLCB has an unconventional phase transition to a fractionalized QSL as a function of temperature. Given the sophistication of the experiments and the wealth of data that they have produced, I would urge the authors to rethink how they have chosen to present their results. The vast overreach is unnecessary.

I have a number of technical issues as well which I will not discuss here since they are minor compared to what I view as the major issue with this manuscript.

Reviewer #3 (Remarks to the Author):

* Main results:

This paper reports the observation of a pressure-induced order-disorder transition in a quantum spin ladder material $C_9H_{18}N_2CuBr_4$. Specific heat and neutron diffraction measurements are conducted. Ising antiferromagnetic (AFM) order is identified at ambient pressure. The AFM transition temperature, excitation gaps of transverse and longitudinal modes, and the Ising order parameter are determined as a function of pressure. All signals show the pressure-induced order-disorder transition consistently. The critical exponents are also estimated. The large critical exponents indicate fractionalization at the quantum phase transition. Quantum Monte Carlo simulation is carried out to calculate the excitation spectrum, and the result is compared with the neutron scattering result. It is claimed that the disordered phase is a Z2 quantum spin liquid with vison and spinon excitations, however, as far as I can see, this claim is not yet supported by solid evidence.

* Significance:

The experimental observation of the pressure-induced order-disorder transition in the spin ladder system is significant.

* Flaws in interpretation/conclusion:

However, the theoretical explanation of the disordered phase as a Z2 spin liquid is not well established in my opinion. The paper provides two arguments for the Z2 spin liquid, one based on the large critical exponent, and the other based on the frustration introduced by DM interaction. Neither of them sounds very convincing to me.

The large critical exponent only indicates fractionalization at the critical point, because the critical exponent is only meaningful at the critical point. Fractionalization at the critical point does not imply fractionalization in the disordered phase. For example, at the deconfined quantum critical point, the spin excitation also fractionalizes, but the spinon confines again in the disordered phase. The experimental observation did not rule out the possibility that the spin ladder simply undergoes a (1+1)D deconfined quantum critical point between the AFM order and the rung dimerization phase [there has been some discussion about this possibility for the spin model in Eq.(1), such as arXiv:1904.00010, arXiv:1904.00021]. The experiment did not try to detect the possible dimer ordering in the high-pressure spin disordered phase, so the nature of the "disordered phase" is unclear yet. Without ruling out the dimer ordering, the claim of spin liquid is not convincing.

The discussion of DM interaction is more speculative. It provides some reasoning for the possibility of the spin-liquid phase, but it can not be treated as evidence for the Z2 topological order. The "broad" excitation spectrum is also not conclusive, because there is no quantitative comparison. How broad is "broad"? Is the Z2 topological order the only possibility to support such a broad excitation spectrum? Why the spectrum broadening can not be simply explained by quantum fluctuations (self-energy corrections) of interacting magnons?

The authors also mentioned by the end of the Quantum Monte Carlo simulation section that "the experimental data cannot be reproduced by QMC calculations for the Hamiltonian in Eq.(1)." This mismatch seems to indicate that the underlying theoretical model is not entirely clear yet, not to mention the correct theoretical understanding of the disordered phase.

I am also confused why the system is treated as a (2+1)D system when the inter-ladder coupling is almost one order of magnitude smaller than the intra-ladder couplings. Shouldn't the system just decouple into (1+1)D spin ladders? For (1+1)D systems, it is not possible to develop topological order, but it is still possible to fractionalize spin excitations by the deconfined quantum criticality.

* Conclusion:

Although the experimental work is impressive, the theoretical explanation is not conclusive yet. The evidence for spin liquid and Z2 topological order is unclear. This prohibits the publication of this paper in its current form. However, I do think the experimental part is worth publication if the theoretical interpretation can be left open, or if conclusive evidence for the Z2 spin liquid could be provided.

We thank all reviewers for their constructive remarks and also their criticisms which certainly help us for improvement of the paper.

Reply to Reviewer #1

In this manuscript, the authors present their comprehensive experimental investigation of the organic magnet DLCB, which is a model $S=1/2$ spin ladder compound. While an isolated ladder has a singlet ground state with a large spin gap, when they are coupled with a large enough interladder coupling, they go into an ordered phase. DLCB is believed to lie just above this quantum critical point with weak magnetic order. The main idea of this work is to tune the magnetic Hamiltonian with hydrostatic pressure and study this quantum phase transition using elastic and inelastic neutron scattering. The authors found that the magnetic order is suppressed with moderate pressure of about 1 GPa. The first important observation is that this quantum phase transition is of second-order nature, which is shown by the continuous suppression of ordered moment size with increasing pressure. From this measurement, they extract the order parameter critical exponent β of 0.68, which is much larger than the value expected for a typical magnetic transition. In addition, the suppression of the longitudinal mode as a function of pressure exhibits power-law behavior with unusually small critical exponent ν . The second observation is the fact that the magnetic excitation spectrum in the phase $P > P_c$ cannot be explained by a conventional picture. If this quantum paramagnet is described by the conventional ladder picture (as assumed in the quantum Monte Carlo study), one would expect a well-defined excitation associated with the singlet-triplet gap. However, the observed spectrum does not show any sharp features and only a rather broad continuum was observed. The authors have carried out a careful examination to rule out any disorder effect for this continuum. The authors go on to argue that this quantum paramagnetic phase is a Z_2 quantum spin liquid with fractionalized excitations, based on the 1.3 GPa spectrum, which seems to show a low-energy vison mode and spinon continuum excitations.

I find this manuscript very interesting and well-written. I think this work will be a valuable addition to the growing list of materials exhibiting quantum phase transitions. In particular, the careful experimental examination of the second-order nature of the transition is impressive and convincing. I recommend publication in Nat. Comm. After the authors address the following concerns:

We are delighted that the referee appreciates the scientific merits of our results, and presentation of our manuscript, and that he/she recommends publication in Nature Communications subject to our consideration of his/her technical comments. We address these below:

1) While I am impressed with the experimental data and presentation about the critical behavior, the experimental evidence for fractionalization is weak. I do agree that the quantum phase transition in this compound seems to be quite unusual. However, I would expect to see much stronger evidence to make the logical jump from that to Z_2 quantum spin liquid. In particular, the vison observation claim relies on only one scan done at the CNCS. The data measured at the same pressure at MACS do not seem to show similar structure. I recommend toning down the fractionalization claim.

We followed the referee's suggestion and toned down the claim about the Z_2 spin liquid phase in the quantum-disordered regime in the revised version of the manuscript. We also want to emphasize that by fractionalization, the Landau order parameter becomes a composite operator of partons, which leads to a large value of anomalous exponent η at the critical point. A large η can explain the broad spectral linewidth of the transverse modes near the transition as shown in Fig. 3(d). As the energy gap of the longitudinal mode (LM) at 1.3 GPa is as small as 0.25 meV, the high-resolution & intensity instrument is necessary to tell the gap from the strong incoherent elastic scattering mainly from the CuBe pressure cell. It explains that the vison/LM was observable by high-resolution neutron data collected at CNCS.

2) Somewhat related comment: I am concerned with seemingly very different spectra from the two instruments. The raw spectra in the Supplementary material do not help either.

In general, INS under high pressure is very time-consuming because of the limited sample space and the heavy attenuation of the neutron beam by the piston-cylinder type pressure cell. As a result, we collected INS neutron data at two different instruments MACS and CNCS in a timely and efficient manner. Their instrumental resolutions have been carefully taken into account to extract the intrinsic linewidth of the spin excitations. Note that we decided to include the raw spectra in the Supplementary Information to show how the background was determined and then subtracted for the data presented in Figs. 3(a) and (b) in the main text.

3) The QMC model calculation was done to show that one specific model for the quantum phase transition (variation of the interladder coupling α) fails to explain the observation. It is not clear to me why just varying α is the best model description.

As shown in the figures below, one may tune the ground state of the coupled $S=1/2$ ladders from the magnetically ordered state to a conventional quantum paramagnetic state by varying either the Ising exchange anisotropy λ or the exchange ratio $\alpha=J_{\text{int}}/J_{\text{leg}}$. In the former case, the gap energy Δ_{TM} of the transverse mode (TM) is expected to become smaller with increase of λ , which is the opposite to experimental observation in Fig. 1 of the main text. Consequently, it can be ruled out. The latter case is supported by analysis of the magnetic interactions under pressure based on the crystal structure as detailed in the Supplementary Information.

Left panel: Ground-state phase diagram of the coupled spin-1/2 ladders as a function of the exchange ratio $\alpha=J_{\text{int}}/J_{\text{leg}}$ and the Ising exchange anisotropy λ . Right panel: Energy gap Δ_{TM} of the transverse mode from QMC and PCUTs as a function of λ for $J_{\text{leg}}=0.6$ meV, $J_{\text{rung}}=0.64$ meV and $J_{\text{inter}}=0.19$ meV. The figures are adapted from Phys. Rev. Lett. **122**, 127201 (2019).

4) The authors go on to discuss various possible microscopic origin for the observed unusual quantum phase transition, including discussions about the DM interactions. However, to me, this discussion is somewhat hollow, since the original Hamiltonian the authors consider, which treats the Ising anisotropy very phenomenologically. That is, λ in the Hamiltonian is assumed to be the same for all three interactions without microscopic justification.

The Ising exchange anisotropy λ originates from the spin-orbit coupling because the orbital angular momentum of Cu^{2+} is not fully quenched in this material. Its detail depends on the local environment of the CuBr_4^{2-} tetrahedra. In our paper, λ is assumed to be same along all J 's to limit the number of fitting parameters. As this anisotropy is weak, it is not possible for our calculation to tell apart the λ 's for the three J 's. This assumption does not affect the main conclusion that for unfrustrated interactions, an ordinary quantum phase transition to a conventional quantum paramagnetic phase emerges thus a mechanism that is not included in the original Hamiltonian is required to interpret the large anomalous exponent η at the critical point and broad magnetic excitation continua near the phase transition.

Reply to Reviewer #2

1. The authors have carried out an impressive study of the heat capacity, magnetic ordering, and magnetic excitations as a function of temperature and pressure in an $S=1/2$ spin-ladder material which they label DLCB. Notably, at a pressure of about 1.0 GPa there is a crossover from Neel order to a quantum spin liquid (QSL) phase.

We are pleased that the referee finds our study impressive. We address his/her technical questions below.

2. Much of the paper is devoted to speculation about the nature of the QSL phase; specifically, the authors argue that it is a gapped QSL with fractionalized degrees of freedom. Indeed, they claim to be able to identify the responses from vison and spinon excitations in the inelastic neutron spectra. Unfortunately, this model for the QSL seems to rely critically on the numerical values of the critical exponents β and ν which are determined from data entirely outside of the critical region. Typically, critical exponents are determined from high precision data in **the reduced temperature** range from 0.1 to 0.001 or 0.0001. Further, the data are analyzed with power laws modified by corrections-to-scaling and the effects of any crossover in universality class. By contrast, the power laws obtained in this paper are the results of fits of the data to simple power laws extending over a broad **temperature** range beginning around 0.1 and extending down to low **temperatures**, that is, entirely outside of the critical region. Thus they represent a parameterization of the low to intermediate **temperature** range data and have nothing to do with critical phenomena and actual critical exponents. This has led the authors to a vast overreach in the interpretation of their otherwise beautiful data. In reality, as far as I can tell, there is no evidence whatsoever that DLCB has an unconventional phase transition to a fractionalized QSL as a function of **temperature**. Given the sophistication of the experiments and the wealth of data that they have produced, I would urge the authors to rethink how they have chosen to present their results. The vast overreach is unnecessary. I have a number of technical issues as well which I will not discuss here since they are minor compared to what I view as the major issue with this manuscript.

As the critical exponents β and ν were extracted from the pressure-dependent neutron data in our paper, we think that the referee meant "the reduced pressure" or "pressure" rather than "the reduced temperature" or "temperature" for the above highlighted words. We don't necessarily agree with his/her statement that the critical exponents should be determined in the reduced control parameter (such as pressure or field or chemical doping) range from 0.1 to 0.001. It ought to be case-dependent on how close the material is to the quantum critical point (QCP) at the ambient condition. For instance, the quantum critical regime of the pressure-induced phase transition in the three-dimensional dimerized antiferromagnet TiCuCl_3 extends over a reduced-pressure range at least up to 1.0 as illustrated in the figures below.

Left panel: Pressure dependence of the measured longitudinal mode in TlCuCl_3 . The black curve was obtained from the theoretical description as $\Delta_L(P) \propto (P-P_c)^{1/2}$, which is valid at least up to the reduced pressure $P_r=(P-P_c)/P_c \approx 1.0$. The figure is adapted from Phys. Rev. Lett. **100**, 205701 (2008); Right panel: Linear proportionality of the measured $T_N(P)$ and $m_s(P)$ in TlCuCl_3 as $I \propto m_s^2(P) \propto T_N^2(P) \propto (P-P_c)$, which is also valid up to $P_r \approx 1.0$. The figure is adapted from Nat. Phys. **10**, 373 (2014).

To address the referee's concern that the critical exponents were determined from data entirely outside of the critical region, we performed a careful examination to estimate the quantum-critical regime in which the critical exponent β can be reliably extracted from the general power law $m \propto (P-P_c)^\beta$. Following the similar procedure as described in the literatures by Nohadani *et al.*, Phys. Rev. B **69**, 220402(R) (2004) and Sebastian *et al.*, Phys. Rev. B **72**, 100404(R) (2005), we plotted β vs. p_w , where p_w is the maximum of the reduced pressure $(P_c-P)/P_c$ of the fit window. The linear extrapolation gives $\beta=0.675$ as $p_w \rightarrow 0$, which is identical to β at $p_w=1$. Consequently, we can conclude that the scaling behavior $m \propto (P-P_c)^\beta$ can be applied for the entire investigated pressure region in DLCB, and so can the scaling behavior $\Delta_L(P) \propto (P_c-P)^{\nu z}$ in the same critical regime. This is consistent with the fact that DLCB is already in the vicinity of the QCP at ambient pressure. We have now included this clarification in the revised version of the Supplementary Information.

Reply to Reviewer #3

* Significance:

The experimental observation of the pressure-induced order-disorder transition in the spin ladder system is significant.

We are pleased that the referee finds our experimental work significant. We address his/her technical questions below.

* Flaws in interpretation/conclusion:

However, the theoretical explanation of the disordered phase as a Z_2 spin liquid is not well established in my opinion. The paper provides two arguments for the Z_2 spin liquid, one based on the large critical exponent, and the other based on the frustration introduced by DM interaction. Neither of them sounds very convincing to me. The large critical exponent only indicates fractionalization at the critical point, because the critical exponent is only meaningful at the critical point. Fractionalization at the critical point does not imply fractionalization in the disordered phase. For example, at the deconfined quantum critical point, the spin excitation also fractionalizes, but the spinon confines again in the disordered phase. The experimental observation did not rule out the possibility that the spin ladder simply undergoes a (1+1)D deconfined quantum critical point between the AFM order and the rung dimerization phase [there has been some discussion about this possibility for the spin model in Eq.(1), such as arXiv:1904.00010, arXiv:1904.00021]. The experiment did not try to detect the possible dimer

ordering in the high-pressure spin disordered phase, so the nature of the "disordered phase" is unclear yet. Without ruling out the dimer ordering, the claim of spin liquid is not convincing.

We thank the referee for bringing the one-dimension (1D) analog of the deconfined quantum critical point (DQCP) to our attention. As pointed out by the referee, the DQCP between the Ising ferromagnetic and valence bond solid (VBS) symmetry-breaking phases in a 1D $S=1/2$ chain system was investigated by numerical methods (arXiv:1904.00010 and arXiv:1904.0002). In case of an $S=1/2$ two-leg ladder, according to the recent numerical study by Ogino *et al.*, Phys. Rev. B **103**, 085117 (2021), the proposed phase diagram as a function of Δ (an exchange anisotropy parameter) and J_4 (a four-spin interaction) is illustrated in the figures below.

Left panel: (a) The rung singlet (RS) phase, (b) the staggered dimer (SD) phase, and (c) the Néel phase in a $S=1/2$ two-leg ladder. Two spins enclosed by dashed line from a singlet. Right panel: Phase diagram of the model on the J_4 - Δ phase with $J_{\text{leg}}=J_{\text{rung}}=1$, where $\Delta>1$ is an Ising exchange anisotropy and J_4 is the four-spin interaction.

As the four-spin interaction is derived from the fourth-order perturbation theory of the Hubbard model, it is expected to be negligible for molecular ladder systems with small exchange interactions (e.g. DLCB). At $J_4=0$, there is a quantum phase transition between the rung singlet (RS) and Néel ordered phases by varying the Ising exchange anisotropy Δ . As the ground state of the rung dimerization phase does not break any symmetry, the RS-Néel transition is not DQCP but instead belongs to the conventional (1+1)D Ising universality class. As a result, the 1D DQCP in DLCB can be excluded. We have now included a discussion about the DQCP in one dimension in the revised version of the manuscript.

The discussion of DM interaction is more speculative. It provides some reasoning for the possibility of the spin-liquid phase, but it can not be treated as evidence for the Z_2 topological order.

We agree with the referee that the possible DM interaction in DLCB cannot be treated as evidence for the Z_2 spin liquid, and therefore soften its consequence in the revised manuscript and deleted the related discussion in the Supplementary Information.

The "broad" excitation spectrum is also not conclusive, because there is no quantitative comparison. How broad is "broad"? Is the Z_2 topological order the only possibility to support such a broad excitation spectrum?

Broad magnetic excitation continua in INS experiments are well-known hallmarks of possible quantum spin liquid states. In the quantum-disordered phase at $P=1.3$ GPa, no surface of the dispersion of the TMs at $T=0.25$ K is observable. The transverse spin excitations form a broad continuum, extending up to 1 meV. Note that the intrinsic spectral lineshape of TMs is significantly (FIVE times) broader than the instrumental resolution. With regard to the possible VBS phase (i.e. the dimer ordering phase), as pointed out by the referee, the spin excitation of VBS fractionalizes only at the deconfined quantum critical point and the spinons become confined

again in the disordered phase. In this respect, the possibility of having the VBS at 1.3 GPa in DLCB can be ruled out.

Why the spectrum broadening cannot be simply explained by quantum fluctuations (self-energy corrections) of interacting magnons?

We agree with the referee that quantum fluctuations of interacting magnons can lead to magnon decay and result in a broad excitation spectrum. For instance, the process of spontaneous magnon decay into two-magnon continuum is allowed only if the following kinematic conditions are satisfied:

$$\mathbf{q}=\mathbf{q}_1+\mathbf{q}_2 \quad (1),$$

$$E_2(\mathbf{q})=E_1(\mathbf{q}_1)+E_1(\mathbf{q}_2) \quad (2),$$

$$E_2(\mathbf{q})^{\min} \leq E_1(\mathbf{q}) \leq E_2(\mathbf{q})^{\max} \quad (3),$$

where $E_1(\mathbf{q})$ is the one-magnon dispersion relation, $E_2(\mathbf{q})^{\min}$ and $E_2(\mathbf{q})^{\max}$ are the lower and upper boundary of the two-magnon continuum, respectively. The first two requirements are conservation of momentum and energy and the last one requires the one-magnon dispersion relation lies between the lower and upper boundary of the two-magnon continuum. As shown in the figure below, in a similar situation to the critical point, the lower boundary of two-particle continuum lies well above the energy gap of TM and therefore the spontaneous decay of TM into a pair of LMs at 1.3 GPa is kinematically forbidden. Moreover, the decay of the TM mode into the three-LM continuum is forbidden due to the energy conservation i.e. three times of Δ_{LM} (0.25 meV) is well above Δ_{TM} (0.56 meV). We have now included this figure and the kinematic conditions under the discussion of ‘‘Spontaneous quasiparticle decays’’ in the revised version of the Supplementary Information.

The authors also mentioned by the end of the Quantum Monte Carlo simulation section that ‘‘the experimental data cannot be reproduced by QMC calculations for the Hamiltonian in Eq.(1).’’ This mismatch seems to indicate that the underlying theoretical model is not entirely clear yet, not to mention the correct theoretical understanding of the disordered phase.

The fact that experimental data around the critical point cannot be reproduced by QMC calculations for the Hamiltonian in Eq. (1) indicates that a mechanism that is not included in the original Hamiltonian is required. We find that the interlayer couplings J_{layer} become more pronounced in the vicinity of the transition as their Cu-Cu separation distances become reduced under pressure. We include a new figure as below showing the magnetic interactions between Cu^{2+} ions in the ac plane, in which the interladder coupling J_{int} and interlayer couplings J_{layer} and J''_{layer} form the triangular spin arrangements. Based on the Goodenough-Kanamori-Anderson rules, all of them are antiferromagnetic exchange interactions. While the coupling J''_{layer} along the a -axis helps to align the ordered moments antiparallel to each other between the adjacent layers in the Néel ordered phase, the coupling J_{layer} can lead to magnetic frustration. One possibility is that the system is driven into an exotic gapped QSL phase with triangular motifs in a three-dimensional interacting spin model. Further theoretical work is certainly necessary to investigate such a scenario.

I am also confused why the system is treated as a (2+1)D system when the inter-ladder coupling is almost one order of magnitude smaller than the intra-ladder couplings. Shouldn't the system just decouple into (1+1)D spin ladders? For (1+1)D systems, it is not possible to develop topological order, but it is still possible to fractionalize spin excitations by the deconfined quantum criticality.

According to QMC calculations, the exchange ratio $\alpha = J_{\text{int}}/J_{\text{leg}}$ is reduced to 14% at the critical point, which is still prominent. As discussed above, we also find that the interlayer couplings J_{layer} become more pronounced near the transition as their Cu-Cu separation distances become reduced under pressure. Overall, the system becomes a three-dimensional ($d=3$) interacting spin framework in the vicinity of the critical point. Moreover, as we have shown above, there is no DQCP for the decoupled $S=1/2$ ladder without the four-spin interactions.

* Conclusion:

Although the experimental work is impressive, the theoretical explanation is not conclusive yet. The evidence for spin liquid and Z_2 topological order is unclear. This prohibits the publication of this paper in its current form. However, I do think the experimental part is worth publication if the theoretical interpretation can be left open, or if conclusive evidence for the Z_2 spin liquid could be provided.

We are pleased that the referee appreciates our experimental results and recommends publication in Nature Communications subject to our consideration of his/her technical comments. We followed the referee's suggestion and toned down the claim about the Z_2 spin liquid phase in the quantum-disordered regime in the revised version of the manuscript.

REVIEWER COMMENTS

Reviewer #1 (Remarks to the Author):

I would like to thank the authors for considering my comments and making changes. Although I appreciate the modifications made by the authors, I still think the overall message has not changed from the previous version. The title still mentions "fractionalization," which in my opinion is not supported (at least unambiguously) by the neutron results. The authors claim that "large anomalous exponent η , which provides evidence for fractionalized degrees of freedom at the critical point." I have an issue with this statement. It is understandable and relevant to discuss the observed critical exponents in the context of the quantum phase transition observed here. However, using the value of η as the *evidence* is probably not justified. Here η is only obtained through the scaling law, which is based on the other two exponents. The correlation length exponent ν is obtained only indirectly through the gap scaling (with a large error bar). In other words, η is only estimated with several theoretical assumptions, and it should not be treated as a piece of solid experimental evidence.

I still believe the experimental results are interesting and worthy of publication, but the current manuscript is still too speculative.

Reviewer #2 (Remarks to the Author):

First, almost all believable measurements of critical exponents are carried out as a function of reduced temperature rather than pressure or some other thermodynamic variable. This is why I began talking about reduced temperature but then neglected to switch to reduced pressure in discussing the experiments at hand. I apologize for any confusion that this might have caused.

One of the most difficult of all challenges in studies of phase transitions is to determine universality classes from apparent critical exponents. The literature is filled with incorrect papers in which the authors attempt to do so by power law fits to data such as the order parameter or susceptibility over some broad parameter range. The power laws invariably work well over one or two decades but the exponents typically describe some crossover effect rather than asymptotic critical behavior. This work most likely falls into that category but, of course, I cannot prove it.

Clearly, the authors want their experiment to be impactful and therefore are committed to their optimistic interpretation of their data. I made this mistake once myself but fortunately discovered the error at a later date and was able to correct it. I predict that these authors will find themselves in the same situation.

I leave it to the editor to decide if the journal wants to publish this article, as is. I would not want to stand in the way in the off chance that my judgement is wrong in this case.

We thank again the referees for the time and effort that they have dedicated to providing the valuable comments. The following paragraphs address the issues raised and describe the implementations in the revised version of the manuscript.

Reply to Reviewer #1

Comments: I would like to thank the authors for considering my comments and making changes. Although I appreciate the modifications made by the authors, I still think the overall message has not changed from the previous version. The title still mentions "fractionalization," which in my opinion is not supported (at least unambiguously) by the neutron results. The authors claim that "large anomalous exponent η , which provides evidence for fractionalized degrees of freedom at the critical point." I have an issue with this statement. It is understandable and relevant to discuss the observed critical exponents in the context of the quantum phase transition observed here. However, using the value of η as the *evidence* is probably not justified. Here η is only obtained through the scaling law, which is based on the other two exponents. The correlation length exponent ν is obtained only indirectly through the gap scaling (with a large error bar). In other words, η is only estimated with several theoretical assumptions, and it should not be treated as a piece of solid experimental evidence.

Reply: We thank the referee for letting us know his/her concern. Please note that Fig. 3(a) in the main text shows the magnetic inelastic scattering of the gapped transverse mode at the antiferromagnetic zone center at $P=1.06$ GPa ($\sim P_c$) after subtracting the background contribution, which was measured at $T=15$ K with the same instrumental configurations. The scattering has a long tail at higher excitation energies, in stark contrast to the predicted line shape (the orange line) of the conventional $S=1$ spin waves. Moreover, such broadening cannot be attributed to the either two-TM or two-LM scattering process because of the energy-momentum conservation rule. A natural explanation of the broadening is owing to fractionalization of the elementary excitations i.e., $S=1/2$ spinons, which are created and detected in pair by neutron scattering. In such a scenario, the magnetic order parameter becomes a composite operator of partons. It should thereby lead to a large value of the anomalous exponent η , which describes the spatial decay of the magnetic correlations. Coldea *et al.*, tried to estimate η from fitting the observed continuum scattering to a generic power law line shape in the frustrated $S=1/2$ triangular antiferromagnet Cs_2CuCl_4 (Phys. Rev. Lett. 86, 1355 and Phys. Rev. B 68, 134424). However, η extracted in such a way may not describe the true quantum critical fluctuations in case of spin GAPPED continuum-like spectra in DLCB.

According to the referee's comment, in the revised version, we added the discussion that the estimate for η , based on the critical exponents β and ν and a (standard) scaling relation, may not represent the true asymptotic value in the quantum critical regime, due to the limitation of instrument resolution. However, even in view of these caveats, the estimated values of η are

certainly significantly larger than the prediction at conventional quantum critical points or its mean-field approximation ($\eta=0$).

Overall, such large values for η , along with the broad spectral linewidth observed near the quantum phase transition, as shown in Fig. 3(d), suggest that the emergence of fractionalized excitations through the pressure-induced quantum phase transition may be realized in DLCB at the critical pressure.

Comments: I still believe the experimental results are interesting and worthy of publication, but the current manuscript is still too speculative.

Reply: We thank the referee for recommending our work for publication. We hope that the above response helps to clarify his/her concerns.

Reply to Reviewer #2

Comments: First, almost all believable measurements of critical exponents are carried out as a function of reduced temperature rather than pressure or some other thermodynamic variable. This is why I began talking about reduced temperature but then neglected to switch to reduced pressure in discussing the experiments at hand. I apologize for any confusion that this might have caused. One of the most difficult of all challenges in studies of phase transitions is to determine universality classes from apparent critical exponents. The literature is filled with incorrect papers in which the authors attempt to do so by power law fits to data such as the order parameter or susceptibility over some broad parameter range. The power laws invariably work well over one or two decades but the exponents typically describe some crossover effect rather than asymptotic critical behavior.

Reply: We thank the referee for clarifying the confusion. To date, a considerable number of theoretical and experimental studies of quantum phase transitions in condensed matter systems have been carried out, including magnetic field-induced Bose-Einstein condensation (BEC) in quantum magnets. The critical exponent ν is associated with the phase boundary of the lower critical field H_c , which separates the quantum paramagnetic from the field-induced canted XY-AFM phases, as $H_c(T)-H_c(0) \propto T^{1/\nu}$. In practice, it involves measurements of the ordering temperature as a function of the temperature-dependent lower critical field $H_c(T)$. The best strategy to estimate the exponent ν of the general power-law is that: determine different values of ν from fitting over different size of the reduced temperature windows and then the critical exponent ν is extrapolated toward $T \rightarrow 0$. For instance, this approach has been successfully implemented to estimate ν that is close to the mean-field value $2/3$ for the BEC universality class in various three-dimensional $S=1/2$ weakly coupled dimer materials such as TiCuCl_3 ,

BaCuSi₂O₆, Ba₃Cr₂O₈, Sr₃Cr₂O₈, and so on. For further details, one may check out the review paper about BEC in quantum magnets by Zapf *et al.* Rev. Mod. Phys. **86**, 563 (2014).

In our analysis, we followed the similar procedure as stated above with the fit window of the reduced pressure down to 0.1 to estimate the critical exponents β and ν . They may not represent the true asymptotic values in the quantum critical regime, due to the limitation of instrument resolution. However, their estimated values are in marked contrast to the prediction at conventional quantum critical points, which suggests an unconventional pressure-induced quantum criticality in DLCB.

Comments: This work most likely falls into that category but, of course, I cannot prove it. Clearly, the authors want their experiment to be impactful and therefore are committed to their optimistic interpretation of their data. I made this mistake once myself but fortunately discovered the error at a later date and was able to correct it. I predict that these authors will find themselves in the same situation. I leave it to the editor to decide if the journal wants to publish this article, as is. I would not want to stand in the way in the off chance that my judgement is wrong in this case.

Reply: We thank the referee for sharing his/her personal story. Of course, we do not want to embarrass ourselves with mistakes. We have been prudent to analyze the data and make conclusions. As pointed out by the first referee, the experimental results are interesting. We hope that by publishing these results, we can promote more experimental and theoretical studies on this intriguing system.

REVIEWERS' COMMENTS

Reviewer #1 (Remarks to the Author):

I would like to thank the authors for making significant changes to the title and the text. I believe that the new nuanced discussion of the critical exponent will be helpful for readers. I recommend the publication of the manuscript as is.

Reply to Reviewer #1

Comments: I would like to thank the authors for making significant changes to the title and the text. I believe that the new nuanced discussion of the critical exponent will be helpful for readers. I recommend the publication of the manuscript as is.

Reply: We thank the referee very much for the constructive comments and recommendation for publication.